# Use of Thermography to Evaluate Alternative Crops for Off-Season in the Cerrado Region

**DOI:** 10.3390/plants12112081

**Published:** 2023-05-24

**Authors:** Alberto do Nascimento Silva, Maria Lucrecia Gerosa Ramos, Walter Quadros Ribeiro Junior, Patrícia Carvalho da Silva, Guilherme Filgueiras Soares, Raphael Augusto das Chagas Noqueli Casari, Carlos Antonio Ferreira de Sousa, Cristiane Andrea de Lima, Charles Cardoso Santana, Antonio Marcos Miranda Silva, Chistina Cleo Vinson

**Affiliations:** 1Faculdade de Agronomia e Medicina Veterinária, Universidade de Brasília, Brasília 70910970, DF, Brazil; 2Empresa Brasileira de Pesquisa Agropecuária, EMBRAPA Cerrados, Planaltina 73310970, DF, Brazil; 3Instituto de Geociências, Universidade de Brasília, Brasília 70910-970, DF, Brazil; 4Empresa Brasileira de Pesquisa Agropecuária, EMBRAPA Meio-Norte, Teresina 64008-780, PI, Brazil; 5Departamento de Engenharia Agrícola, Universidade de Viçosa, Viçosa 36570900, MG, Brazil; 6Departamento de Ciência do Solo, Universidade de São Paulo, Piracicaba 13418900, SP, Brazil

**Keywords:** gas exchange, thermography, *Amaranthus cruentus*, *Chenopodium quinoa*, *Fagopyrum esculentum*, *Phaseolus vulgaris*

## Abstract

Future predictions due to climate change are of decreases in rainfall and longer drought periods. The search for new tolerant crops is an important strategy. The objective of this study was to evaluate the effect of water stress on the physiology and productivity of crops with potential for growing in the off-season period in the Cerrado, and evaluate correlations with the temperature of the canopy obtained by means of thermography. The experiment was conducted under field conditions, with experimental design in randomized blocks, in a split-plot scheme and four replications. The plots were: common bean (*Phaseolus vulgaris)*; amaranth (*Amaranthus cruentus*); quinoa (*Chenopodium quinoa*); and buckwheat (*Fagopyrum esculentum*). The subplots were composed of four water regimes: maximum water regime (WR 535 mm), high-availability regime (WR 410 mm), off-season water regime (WR 304 mm) and severe water regime (WR 187 mm). Under WR 304 mm, the internal concentration of CO_2_ and photosynthesis were reduced by less than 10% in amaranth. Common bean and buckwheat reduced 85% in photosynthesis. The reduction in water availability increased the canopy temperature in the four crops and, in general, common bean was the most sensitive species, while quinoa had the lowest canopy temperatures. Furthermore, canopy temperature correlated negatively with grain yield, biomass yield and gas exchange across all plant species, thus thermal imaging of the canopy represents a promising tool for monitoring crop productivity for farmers, For the identification of crops with high water use management for research.

## 1. Introduction

In Brazil, the Cerrado region has three crop-growing periods: (1) the main crop (period with the highest rainfall), from October to January; (2) the second crop, at the end of rainy season, or off-season, from February to May, with less rainfall available; and (3) the winter crop, almost without precipitation, grown using irrigation from May to September. In the second harvest, corn (*Zea mays*) and common bean (*Phaseolus vulgaris*) are predominant [1]. However, the literature shows the high susceptibility of both crops to water deficit that causes a reduction in their productivity [2,3]. Thus, the study of different crops to evaluate the most tolerant crops to periods of water deficiency, as well as the knowledge of the plant mechanisms used to tolerate prolonged periods of drought, is useful for the maintenance of agricultural production in the Cerrado during the off-season period.

The species that stand out with potential to be alternative crops during the off-season period are quinoa (*Chenopodium quinoa*), amaranth (*Amaranthus cruentus*) and buckwheat (*Fagopyrum esculentum*). Quinoa is a pseudocereal, originally from the Andes region, which is well-adapted to abiotic stresses, such as water scarcity, low temperatures, salinity and nutrients-poor soil [4,5]. Its grains contain unsaturated fatty acids, antioxidants, essential amino acids and high levels of gluten-free proteins, and are rich in Fe, Mg, fibres and vitamins [6,7,8,9]. Amaranth is a species widely cultivated in the Andes. It produces grains with a high nutritional value, flavour and attractive colour. Its leaves have also been widely used as human and animal food [10,11]. Buckwheat is a species belonging to the Polygonaceae family, also known as black wheat. It has the potential to be used as a green manure, cover plant and animal feed [12], and Ha recently been consumed by humans in the form of pasta in Japan, Italy and China [13]. Historically, seeds have been widely used as flour in bread manufacture and in the production of vinegar and tea [14]. Like amaranth and quinoa, it presents itself as an alternative for people with celiac disease, due to the absence of gluten in its composition [15]. For the assessment of the potential of different species to grow in the off-season period, an investigation of their physiological responses to water stress is essential to understand their drought tolerance mechanisms.

Plants have several mechanisms of protection in response to water deficit. One of the first reactions of the plant is stomatal closure, which reduces water loss via transpiration and, ultimately, photosynthesis [16]. Another mechanism is the reduction in leaf area by decreasing cell expansion [17]. In order to maintain the water potential and cell turgor, some species accumulate low-molecular-weight osmotically active substances in the cytosol, such as proline [18], maintaining the stomatal opening and CO_2_ uptake, even if the plants are kept under low soil water potential [19]. Proline is a widely studied solute in crops, due to its highly sensitive response to stress conditions [20,21]. Proline has osmoregulatory and osmoprotectant characteristics, acting in the maintenance of integrity of proteins and membranes, has an antioxidant effect playing a role in the removal of reactive oxygen species and participates in molecular signalling for the expression of specific genes related to stress [22].

Under field conditions, the yield and physiological evaluations are essential for phenotyping and selecting the plants tolerant to water stress [23]. However, traditional methods have limitations, such as time, labour, equipment and costs [24]. To overcome such limitations, methods such as non-invasive imaging techniques have been developed to complement or even replace traditional methods [25,26]. Thermal imaging has been widely used to characterize the plants subjected to water stress [27,28,29]. In recent years, canopy temperature has been evaluated through thermal cameras coupled in unmanned aerial vehicles (UAVs) [30], which is able to assess several genotypes or different crops at the same time under different levels of water stress, and the data obtained are related with stomatal opening, biomass production and water use efficiency [31,32]. It is known that leaf temperature is altered via stomatal opening and transpiration rate: the temperature rises as the stomata closes under water stress [33]. In addition, canopy temperature is related to leaf transpiration cooling and water relations in plants [31]. Additionally, it has been suggested that a lower canopy temperature increases water use efficiency and above-ground biomass production, in addition to productivity [32]. Several authors have found correlations between canopy temperature and physiological parameters, such as photosynthesis and transpiration rates and stomatal conductance, for several species such as rice, soybean, maize and wheat [34,35,36,37]. Those traits can validate the thermal measurements as the selection criteria for drought.

We hypothesized that water stress alters plant physiology and productivity, and is correlated with the temperature of the canopy of crops for cultivation in the off-season period, which can be used as a tool in breeding programs.

Thus, the objective of this work was to evaluate the effect of water stress on the physiology and productivity of crops with the potential for cultivation in the off-season period in the Cerrado, and investigate the potential correlations with the temperature of the canopy obtained through thermography.

## 2. Results

### 2.1. Physiological Variables

The results of this study demonstrated a significant effect of the different water regimes on all the studied crops (Table 1). When comparing the maximum water regime with the severe water regime, we detected reductions of 42, 81, 70 and 82% in net assimilation of CO_2_; 40, 93, 90 and 92% in stomatal conductance; NS, 53, 56 and 41% in internal CO_2_ concentration and 33,87, 79 and 82% in transpiration for amaranth, common bean, quinoa and buckwheat, respectively. 

Water stress did not have a significant influence on water use efficiency for the four studied crops. The reduction in water availability from the maximum water regime to the off-season water regime caused reductions of 8, 67, 13 and 68% in the net assimilation of CO_2_; 10, 89, 47, and 87% in stomatal conductance; NS, 31, NS and 58% in the internal concentration of CO_2_ and NS, 77, 28 and 72% in transpiration for amaranth, common bean, quinoa and buckwheat, respectively (Table 1).

In the comparison between species, under conditions of maximum and high-availability regime, amaranth presented the highest rate of net photosynthesis (approximately 44 µmol CO_2_ m^−2^ s^−1^), whereas quinoa showed the highest stomatal conductance (approximately 0.74 µmol CO_2_ m^−2^ s^−1^), internal concentration of CO_2_ (approximately 280 µmol CO_2_ m^−2^ s^−1^) and transpiration (approximately 12 µmol H_2_O m^−2^ s^−1^) (Table 1). Under the off-season water regime, amaranth showed the highest photosynthesis rate (40 µmol CO_2_ m^−2^ s^−1^) and transpiration rate (7.5 µmol H_2_O m^−2^ s^−1^). Under conditions of severe water stress, amaranth was the species with the highest rates, with a net photosynthesis of 25 µmol CO_2_ m^−2^ s^−1^, stomatal conductance of 0.18 µmol CO_2_ m^−2^ s^−1^ and transpiration of 5.3 µmol H_2_O m^−2^ s^−1^. Regarding water use efficiency, amaranth was the most efficient species, both under the conditions of the maximum water availability regime (5.8 µmol of CO_2_ for each molecule of water) and severe water stress (4.7 µmol of CO_2_ for each water molecule).

The gas exchange data reveal differences in the responses of the species to progressive water restriction. Amaranth and quinoa maintained gas exchange and photosynthesis until the off-season regime, with major effects only observed under the severe regime (A, gs and E, Table 1). Common bean, on the other hand, was more sensitive, with water availability under the off-season regime proving inadequate for the maintenance of gas exchange (A, gs and E, Table 1). Finally, buckwheat showed differences between the maximum and high regimes, with gradual decreases from the maximum water regime to the severe regime (A, gs and E, Table 1).

The average values of the chlorophyll indexes (*a*, *b* and total) and concentration of free proline in the leaves are shown on Table 2. For chlorophyll *a*, the severe water regime caused reductions of 33 and 35% in amaranth and common bean, respectively. A similar result was observed in the chlorophyll *b* and total chlorophyll indexes. When comparing the maximum water regime with the off-season water regime, there were no reductions in the levels of chlorophyll *a*, chlorophyll *b* and total chlorophyll in any of the studied crops. In general, quinoa had the highest levels of chlorophyll *b* and total chlorophyll. In fact, the concentration of total chlorophyll in quinoa increased under the severe regime.

Proline accumulation in the leaves of the plants exposed to drought stress was a sensitive analysis. The severe drought promoted an increase in proline content of 104, 54 and 196% in amaranth, common bean and quinoa, respectively (Table 2). The severe drought promoted an increase in proline content of 104, 54 and 196% in amaranth, common bean and quinoa, respectively (Table 2), but for buckwheat, the proline content increased only by 15% under severe water stress.

Under the off-season water regime, increases of 53, 50 and 143% were measured for amaranth, common bean and quinoa, respectively. Under the maximum and high-availability water regimes, all the studied crops showed similar concentrations of proline in the leaves, whereas under the lower water regimes (off-season and severe), amaranth and quinoa were the species with the highest concentrations of proline (Table 2).

### 2.2. Productivity of Dry Biomass and Grains

The results from dry biomass, grain yield and productivity per unit of water applied are shown in Table 3. In general, quinoa and buckwheat showed significant reductions in dry biomass only under the severe water regime, whilst amaranth and common bean reduced their dry biomass under the off-season and severe water regimes. The water deficit reduced the dry biomass in all cultures and, when comparing the maximum water regime with the severe water regime, reductions of 80, 83,67 and 58% were observed for amaranth, common bean, quinoa and buckwheat, respectively. When comparing the maximum water regime with the off-season water regime, reductions in dry biomass of 43 and 31% were observed for amaranth and common bean, respectively, whilst quinoa and buckwheat were unaffected. Under severe water stress conditions, quinoa, buckwheat and amaranth produced similarly low dry biomasses (between 3.490 and 4.878 kg ha^−1^), and common bean produced even less than that, with 1.162 kg ha^−1^. Under the off-season water regime, quinoa was the species that produced the greatest amount of dry biomass (15.881 kg ha^−1^). Of note, however, the severe water regime applied in this study, simulating only 187 mm of rain, very rarely to occurs during the off-season period in the Cerrado (February and April). Generally, the average values of precipitation are observed close to or above 300 mm, which corresponds to the off-season water regime applied in this study (Table 3).

Grain yields of up to 3545, 5.295, 3.622 and 2.387 kg ha^−1^ were observed for amaranth, common bean, quinoa and buckwheat, respectively (Table 3). However, water stress reduced the grain yield of the four studied crops. When comparing the maximum water regime with the severe water regime, there were reductions of 84, 91, 87 and 70% for amaranth, common bean, quinoa and buckwheat, respectively, whilst common bean was extremely sensitive, with a yield loss of 3685 kg ha^−1^. Under the off-season water regime, the species that stood out were amaranth (2724 kg ha^−1^) and quinoa (3266 kg ha^−1^), which also showed no significant reduction compared to the maximum regime.

Regarding productivity per unit of available water (PUAD) in general, water stress reduced the weight of grains per millimetre of water applied (Table 3). Under the conditions of severe water stress, all the studied crops showed similar values of PUAD. Under the conditions of maximum water regime, common bean and quinoa had the highest PUAD (9.89 and 7.90 kg ha mm^−1^). Under the off-season water regime, amaranth and quinoa were the species with the highest PUAD (9.00 kg mm^−1^ and 10.88 kg mm^−1^, respectively).

### 2.3. Average Canopy Temperature

The average temperature of the canopy of amaranth, common bean, quinoa and buckwheat under the four water regimes is shown in Table 4. In general, the reduction in water availability increased the temperature of the studied crops. Under the conditions of severe water stress (187 mm), there were increases of 18, 36, 20 and 16% in relation to the maximum water regime (535 mm) for amaranth, common bean, quinoa and buckwheat, respectively. Comparing the maximum and high regimes with the off-season water regime, the only significant increase in temperature was observed for common bean when compared to the high water regime, whilst the other three species were not significantly affected. Common bean was the species with the highest canopy temperature both under the condition of sever water regime and the off-season regime (34.5 and 28.5 °C, respectively). Quinoa was the species that, in general, had the lowest temperatures, either under the conditions of severe or off-season water regimes.

### 2.4. Pearson’s Correlation

The correlation between the physiological, productive and average canopy temperature variables are shown in (Figure 1) for common bean (Figure 1A), quinoa (Figure 1B), amaranth (Figure 1C) and buckwheat (Figure 1D). It is noteworthy that a correlation is considered weak at 0.00 < ρ < 0.30, moderate at 0.30 ≤ ρ < 0.60, strong at 0.60 ≤ ρ < 0.90 and very strong and linear at 0.90 ≤ ρ < 1.00 (Callegari-Jacques, 2003).

For the common bean crop, a very strong correlation was observed between photosynthesis and grain yield (0.98), photosynthesis and biomass productivity (0.98), and grain yield and biomass yield (0.98). Strong and very strong correlations were also observed between grain yield and transpiration (0.91), grain and gas productivity (0.98), grain yield and chlorophyll *b* (0.75), grain yield and chlorophyll *a* (0.73) and grain yield and internal CO_2_ concentration (0.63) (Figure 1). The average temperature of the canopy was negatively correlated with almost all the variables studied, except for proline (0.36). There were strong and very strong negative correlations of canopy temperature with chlorophyll *b* (−0.91), chlorophyll *a* (−0.9), photosynthesis (−0.89), biomass production (−0.89) standout) and grain yield (−0.89). Moreover, proline was negatively correlated with all variables, except canopy temperature. However, such correlations were considered moderate and weak (<0.6).

In the quinoa culture, the highest magnitudes of positive correlation for grain yield were observed with stomatal conductance (0.96) and transpiration (0.96). There was also a very strong positive correlation between grain yield and the internal concentration of CO_2_ (0.91), biomass production (0.9) and photosynthesis (0.85). The average temperature of the canopy was negatively correlated with almost all the variables studied, except for proline (0.76), in a similar way as the common bean culture. Importantly, we detected strong negative correlations between canopy temperature and transpiration (−0.92), biomass production (−0.91), internal CO_2_ concentration (0.90), stomatal conductance (−0.88) and grain yield (−0.87). Proline was negatively correlated with almost all variables, except for canopy temperature and *a*, *b* and total chlorophyll. However, unlike what occurred in the common bean culture, these correlations were considered strong and very strong (>0.6), with an emphasis on the correlations between proline and stomatal conductance (−0.92), grain productivity (−0.87), concentration internal CO_2_ (−0.85) and transpiration (−0.84).

For amaranth, the grain yield was positively correlated with all variables, except canopy temperature. Correlations of grain yield were observed with biomass yield (0.91), 0.84 with chlorophyll *a*, 0.79 with chlorophyll *b*, 0.81 with stomatal conductance, 0.82 with photosynthesis and 0.7 with transpiration. The mean temperature of the canopy negatively correlated with all variables, except the internal concentration of CO_2_ (0.46) and proline (0.68). Negative correlations stand out between the average canopy temperature and total chlorophyll (−0.91), chlorophyll *a* (−0.9), grain yield (−0.89), chlorophyll *b* (−0.89), photosynthesis (−0.82), stomatal conductance (−0.81) and biomass productivity (−0.81). Proline was also negatively correlated with all variables, except for the internal CO_2_ concentration. However, such correlations were considered moderate and weak (<0.6).

In the buckwheat culture, the highest magnitudes of positive correlation were observed between stomatal conductance and photosynthesis (0.95), stomatal conductance and transpiration (0.96), and photosynthesis and transpiration (0.96). Positive correlations between grain yield and biomass yield (0.87), internal CO_2_ concentration (0.42), stomatal conductance (0.7), photosynthesis (0.79) and transpiration (0.75) were observed. The average canopy temperature was negatively correlated with almost all the variables, except chlorophyll b (0.17), total chlorophyll (0.59) and proline (0.33). There was a strong negative correlation of this variable with grain yield (−0.87) and biomass (0.85). Unlike what occurred in the other species, chlorophyll *a*, *b* and total chlorophyll correlated negatively with almost all the variables studied, and proline correlated positively with most variables, except grain and biomass productivity.

Pearson’s correlation for amaranth and common bean, in general, were the species with strong negative correlations between canopy temperature and chlorophyll *a*, *b* and total chlorophyll. On the other hand, for quinoa and buckwheat, there was a positive correlation between these variables, a weak and strong correlation, respectively.

## 3. Discussion

### 3.1. Physiological Variables

Our results indicate that under the conditions of water stress, physiological changes occur to decrease water loss and prevent plant death [38,39], and these changes depend on the intensity of the stress, the species and the stage of development of the plant [40]. In general, there was a reduction in gas exchange (net assimilation rate of CO_2_, stomatal conductance, internal concentration of CO_2_ and transpiration) under both the severe and off-season water regime conditions, a water condition commonly found during the off-season period in the Cerrado (Table 1).

One of the first physiological responses of plants to water stress is stomatal closure, which is an attempt to maintain the water content in plant tissues for a longer period [16]. When the stomata are open, CO_2_ is assimilated, and H_2_O is lost through transpiration [41]. However, when the water supply becomes inadequate, the stomata close, affecting the uptake of CO_2_ and transpiration [41,42]. When analysing the gas exchange among the study crops (photosynthesis and stomatal conductance), amaranth showed the lowest reduction due to drought, followed by quinoa, while the common bean and buckwheat were the most sensitive.

Our study showed that amaranth had the highest photosynthesis rate under all the studied water regimes (Table 1). Such results may be associated with the photosynthesis mechanism of each species, as amaranth is a C_4_ plant [43], whereas the common bean, quinoa and buckwheat are C_3_ plants. C_4_ plants have a much higher rate of net photosynthesis than C_3_ plant species, because photorespiration is practically null and the high affinity of the enzyme phosphoenolpyruvate carboxylase for the substrate (CO_2_) allows for photosynthesis in C_4_ plants to occur with a reduced stomatal opening and consequently, with a low water loss [44,45].

Amaranth was the least sensitive to water stress under all water conditions, as the percentages of reduction in gas exchange variables were much lower than those observed for the least sensitive species, while common bean and buckwheat were the most sensitive, with reductions greater than 80% in net CO_2_ assimilation, stomatal conductance and transpiration. Again, the high susceptibility of common bean and buckwheat to water stress stands out, as there were reductions greater than 60% in the net assimilation of CO_2_, and greater than 85% in the stomatal conductance, in addition to reductions in the internal concentration of CO_2_ and transpiration.

Amaranth is tolerant to heat and drought and resistant to pests and diseases [46]. The fact that amaranth is a C_4_ plant increases the tolerance in photosynthesis efficiency in hot and dry environments, with this species being able to produce a high amount of biomass with low water consumption [43]. Amaranth’s response to water stress was reduced by up to 44.48% in the net CO_2_ assimilation rate, which ranged from 15.10 to 27.06 µmol CO_2_ m^−2^ s^−1^; similar results found in this study [47]. The rapid development of the root system allows for the absorption of water in the deeper layers of the soil, combined with the partial closure of the stomata to maintain the internal water status in the plant, are the strategies used by amaranth to increase drought tolerance [43].

On the other hand, the common bean crop has been shown to be quite sensitive to water stress [18], despite being traditionally grown during the off-season period in the Cerrado [48], and it has a shallow root system [49]. Chlorophyll (*a*, *b* and total) is a physiological attribute usually affected under the conditions of water stress. Here, there was a reduction in chlorophyll indexes (*a*, *b* and total) only in amaranth and common bean cultures when grown under severe water stress conditions. It is noteworthy that such reductions could have directly influenced the decrease in the photosynthesis rate in the common bean crop. Reductions in chlorophyll *a*, *b* and total concentrations in common bean genotypes subjected to water stress were also observed in the literature (i.e., [40,50]). Reductions in leaf pigments induced by drought are considered an indicator of oxidative stress, which is attributed to pigment photo-oxidation, chlorophyll degradation and/or decreased synthesis [51]. Under water stress, there is an increase in the production of reactive oxygen species, increasing lipid peroxidation, and consequently the destruction of chlorophyll [52].

Proline is a widely studied solute in commercial crops, due to its highly sensitive response to stress conditions [20,21]. It has an osmoregulatory characteristic, preventing hyperosmotic stress, balancing the differences in the concentration between the cytoplasm and the central vacuole of the plant cells; an osmoprotective characteristic, acting on the integrity of proteins, enzymes and membranes; an antioxidant function by removing reactive oxygen species; and participates in signalling events for the expression of specific stress genes [22]. In addition, it acts as a storage for carbon and nitrogen for a post-stress period [53,54,55]. The increase in water deficit, both during the off-season and under severe water stress, increased the concentration of proline in the leaves of amaranth, quinoa and common bean. A similar increase in proline under water stress was found in rice [35]. However, proline did not have an osmoregulatory and osmotic adaptation role in these three cultures, since it was weakly and negatively correlated with almost all the physiological and productive variables (Figure 1). Similar results were obtained by [56], who observed that water stress significantly increased the production of free proline in the amaranth species; however, they did not find a positive correlation between the production of proline and the maintenance of the turgor or leaf area. According to the authors, this result indicated that proline did not play a role in osmoregulation or osmotic adaptation, but only as a reserve of carbon and nitrogen during water stress. Gomes et al. [39] also did not obtain an association of proline with osmoregulation, but instead with the preservation of the cell membrane of plants.

The ratio between the photosynthesis rate and transpiration rate results in water use efficiency, i.e., WUE, a physiological parameter that expresses gas exchange in the leaf. In the present study, water stress did not alter the WUE in the studied crops. However, generally, plants under water stress conditions have an increase in resistance to diffusion of CO_2_ in the mesophile, decreasing the efficiency of carboxylation, and therefore have lower WUE [47,57]. Plants with limited water restriction show an increase in the WUE, since the partial reduction in stomatal opening limits transpiration more than CO_2_ entry, increasing the WUE [58]. In the comparison between the studied cultures, we observed that amaranth, both under maximum and severe water availability (5.80 and 4.73 µmol of CO_2_ for each water molecule, respectively), was the most efficient species in terms of WUE. Similar results for amaranth were also found by Valdayskikh et al. [43], which is, according to the authors, indicative of the adaptive properties of amaranth to water stress, such as adequate tissue hydration, stomatal closure and the reduction in transpiration.

### 3.2. Production of Dry Matter and Grain

The grain and biomass productivity of the four studied crops showed similar results to those observed for the physiological variables, corroborating with the high correlation between these variables, as shown in Figure 1. The results of grain productivity under the off-season water regime shows that amaranth and quinoa were the species with the highest yields (2724 and 3266 kg ha^−1^, respectively), and common bean and buckwheat showed the lowest yields (1613 and 1892 kg ha^−1^, respectively). Common bean was extremely sensitive to water stress, as there was a reduction of 70% in its productivity when cultivated under the off-season water regime, a drastic reduction with a possible huge economic loss for farmers cultivating this crop, whilst for amaranth, quinoa and buckwheat, the reduction was much smaller (21, 20 and 23%, respectively). Quinoa and amaranth are therefore excellent alternatives to common bean for growing during the off-season period, as they are superior in grain productivity and produce a greater amount of dry biomass. This fact is relevant, as this will provide greater soil protection for the main season.

The reduction in grain yield and dry biomass due to the effects of water stress on common bean plants has been reported in several studies [16,40,59], with reductions of up to 60% in grain yield of common bean genotypes subjected to water stress [50]. During the crop cycle, the common bean in most cultivation is about 100–120 days, resulting in approximately 400 mm of water required under suitable growing conditions [59]. Assuming that the approximate 400 mm of water would be needed to meet the crop demand in this experiment, the decrease of only 96 mm (off-season water regime) resulted in a reduction of approximately 70% in the productivity of this culture, hence the high susceptibility of common bean to water stress in this work. 

### 3.3. Average Canopy Temperature

All studied species showed strong negative correlation between canopy temperature and gas exchange (stomatal conductance, photosynthesis and leaf transpiration). Water stress causes a stomatal closure and as a consequence of the decreased photosynthesis, leading to an increase in canopy temperature [25]. Although the reduction in water loss, caused by stomatal closure, may pose an immediate advantage to prevent the dehydration of the tissue, because of the decreased transpiration, the leaf temperature increases whilst decreasing the photosynthesis rate to levels insufficient to replace the carbon used as a substrate in respiratory processes [60].

Consequently, this causes a reduction in crop productivity, as observed in the present study. Here, water stress reduced the stomatal conductance, causing increases in the temperature of the canopy (Table 3 and Table 4).

An increase in canopy temperature strongly influenced the grain yield and plant biomass of the four studied crops. Similar results were obtained for other crops, such as rice and canola [35,61]. The highest water regimes (535 mm and 410 m), in general, showed the lowest canopy temperature for the studied crops, and presented the highest plant biomass and grain yield. In addition, cooler temperatures may also increase the WUE in plants, as obtained by Sexton et al. [32].

For amaranth, quinoa and buckwheat, there was no considerable increase in temperature when grown under the off-season water regime (410 mm), and the reductions in the grain yield of these crops were lower than those observed in the common bean crop. This increase in common bean canopy temperature may be related to the high reductions in stomatal conductance and transpiration (Table 1). Under the conditions of severe water stress, there were significant increases in canopy temperature in the four studied crops. C_3_ plants, such as common bean, are more affected by the increase in leaf temperature than C_4_ plants, because in such conditions, the rates of photorespiration increase faster than those of photosynthesis [45].

When the leaf temperature is increased, the rate of photorespiration increases considerably in C_3_ plants, reducing net photosynthesis. Due to the CO_2_ concentration mechanism, C_4_ plants reduce the photorespiration rate to negligible levels, even at high temperatures [45,62].

Thermography indicated that the decrease in leaf transpiration contributed to the increase in the temperature of the canopy, which was due to the higher concentration of energy in the form of latent heat, making the leaf temperature higher than the air temperature. The use of thermography revealed that canopy temperature was strongly negatively correlated with grain yield. Thus, under the severe water regime, plants responded physiologically through stomatal closure and increases in leaf temperature, which makes this response to the environment an excellent indicator of the water status of the culture at any given time, and shows that plant canopy temperature is a fundamentally important variable in the assessment and monitoring of water stress [63]. Thermal imaging of the canopy therefore represents a promising tool for the remote monitoring of crop productivity in terms of water use stress management [64]. In addition, thermal imaging can be used remotely, and it is possible to measure large areas, as plants under water stress close the stomata and increases the leaf temperature stress [65].

Proline content in the leaves showed a moderate correlation with canopy temperature for common bean and buckwheat (0.38 and 0.47, respectively). On the other hand, a strong correlation was obtained for amaranth and quinoa (0.68 and 0.76, respectively). These results may indicate that these species probably have different osmotic mechanisms under water stress, and the osmotic adjustment promoted by increasing the proline content may be less related to canopy temperature than gas exchange.

In relation to chlorophyll *a*, *b* and total chlorophyll, common bean and amaranth presented a strong negative correlation with canopy temperature, which could indicate the chlorophyll degradation, pigment photo-oxidation and a lower synthesis of chlorophyll [66].

## 4. Material and Methods

### 4.1. Experimental Design

The experiment was conducted under field conditions at the Embrapa Cerrados field station, in Planaltina, DF, Brazil (latitude 15°35′30″ S, longitude 47°42′30″ W) (Figure 2), between June and October 2017. The choice of this period for the implantation of the experiment, different to the traditional off-season period, is justified by the low rainfall (Figure 3), making the accurate application of water regimes possible through irrigation.

The region climate, according to the Köppen classification, is of the Aw type [67], with dry winters and rainy summers. The average annual precipitation is 1400 mm, and the average annual temperature is 21.3 °C. The average precipitation and temperature data for the last 20 years (1997–2016) and 2017 (year of the experiment) are shown in Figure 3.

The soil is classified as Oxisol [68] and has the following characteristics in the 0–20 cm layer: pH (water) = 5.77; Ca (mg kg^−1^) = 669.34; Mg (mg kg^−1^) = 171.40; K (mg kg^−1^) = 207.55; H+Al (mg kg^−1^) = 399.76; P (mg kg^−1^) = 48.56; S (mg kg^−1^) = 19.71; and organic matter = 26 g kg^−1^.

The history of the last ten years of cultivation of the experimental area is shown in Table 5. Before the cultivation of soybeans, during the 2005/2006 harvest, the area was under pasture for a long period.

A randomized block design was used in a split-plot scheme, with four replications. The plots were composed of the following: common bean (*Phaseolus vulgaris)*, cultivar BRS Highce; amaranth (*Amaranthus cruentus*); quinoa (*Chenopodium quinoa* “genotype derived from BRS Piabiru”); and buckwheat (*Fagopyrum esculentum*). The subplots were composed of four water regimes: maximum water regime (WR 535 mm), high-availability regime (WR 410 mm), off-season water regime (WR 304 mm) and severe water regime (WR 187 mm). The off-season water regime of 304 mm is a water condition that is common during the off-season in the Cerrado (Figure 3). The morphological, productive characteristics and the photosynthesis mechanism of the studied species are described in Table 6.

The crops were sown in the second week of June 2017, under no-tillage. A spacing of 0.5 m between the rows was used for all crops, and a sowing density of 14 plants m^−1^ for common bean (10 plants m^−1^ for amaranth, 20 plants m^−1^ for quinoa and 70 plants m^−1^ for buckwheat). Fertilization at planting was carried out in the furrows at a dose of 400 kg ha^−1^ of the formula 04-30-16, totalling 16 kg ha^−1^ of N, 120 kg ha^−1^ of P_2_O_5_ and 64 kg ha^−1^ K_2_O. Cover nitrogen fertilization was applied 30 days after planting at 100 kg ha^−1^ N, in the form of urea. In order to avoid competition from invasive plants, manual weeding was carried out.

The different water regimes were obtained using a sprinkler irrigation bar with two sides, each 20 m in length, connected to a self-propelled car with adjustable speed. In this experiment, only one side of the bar was used. During the first 35 days after emergence (DAE), the irrigation of the cover crops and common bean was uniform, and the accumulated level of irrigation was 135 mm (Figure 4A). After this period, the line source methodology was adapted [69] using sprinklers with decreasing flows, from the central area to the end of the irrigation bar [70], as shown in Figure 4. The overlap between the sprinklers produced a gradient of water from the centre to the end of the irrigation bar. To measure the amount of water applied in each irrigation, two rows of collectors were installed parallel to the side of the bar, to measure the volume of the water applied.

During this phase, under different irrigation depths, eleven irrigations were carried out. Along the side of the irrigation bar, four sub-plots were defined, four meters long each, and containing eight cultivation lines spaced 0.5 m apart, representing the four water regimes (535, 410, 304 and 187 mm), which were spaced at 4, 8, 12 and 16 m from the beginning of the irrigation bar to its end, respectively (Figure 4A). The greatest water regime (maximum water regime, 535 mm) was determined as described in the irrigation monitoring program in the Cerrado [71], using wheat culture, the region’s agrometeorological indicators, the type of soil and the date of germination as a reference. The data of soil moisture under the four studied water regimes, at the depths of 0–5, 5–10 and 10–20 cm are shown in Figure 4B.

It is noteworthy that, although the assessments were made up to the 187 mm water regime, it is rare for precipitation to occur at such low levels during the entire off-season period [72]. Generally, values equal to or greater than 300 mm are observed in the most productive regions in the Cerrado biome, which is close to the second water regime applied in this work (304 mm) [72]. However, evaluations were made up to the 187 mm regime to verify the species’ behaviour under extreme water stress conditions. Although this experiment is composed of different species, each with different water requirements, for practicality, we use wheat as a reference, understanding that common bean, buckwheat, amaranth and quinoa have a water consumption similar to that of wheat. The irrigations were carried out with a watering shift of approximately five days, according to the climatic conditions and the phenological phase of the crop.

### 4.2. Variables Analysed

During the flowering period of the crops, 30 days after the beginning of the imposition of the water regimes, the following variables were evaluated: proline content in the leaf (Pr), chlorophyll (*a*, *b*, total), gas exchange (photosynthesis (A); stomatal conductance (gs), internal CO_2_ concentration in the leaves (Ci); and transpiration (E)), and water use efficiency (WUE). The productivity of dry biomass and grains, and the temperature of the canopy were also evaluated. All physiological sampling was done on the youngest, fully expanded leaves.

The samples of the leaves of three plants per plot were collected in the afternoon, between 13:30 and 15:00 af flowering stage, and immediately placed in liquid nitrogen and kept in a freezer at −80 °C for the measurement of proline content. The proline content was determined employing a colorimetric method, according to [73]. The sample extract was obtained by macerating 0.5 g of fresh leaf matter in 10 mL of 3% sulfosalicylic acid, and the optical density was evaluated at 515 nm, using a Pharmacia Ultrospect III spectrophotometer. The absorbance obtained was compared with the standard curve of purified proline (d-proline, Sigma-Aldrich, Saint Louis, MO, USA), and the results expressed in µ mol proline g^−1^ of fresh matter.

The chlorophyll index (*a*, *b*, and *a* + *b*) was evaluated in the morning, on the same day as the photosynthesis evaluation, with a digital ChlorofiLOG, model CFL 1030, by Falker, with ten repetitions for each subplot.

Gas exchange was measured using an IRGA infrared gas analyser model LI-6400XT (LI-COR, Inc., Lincoln, NE, USA). To determine the light saturation, a photosynthetically active photon flux density curve (PPFD) of 0, 20, 60, 100, 250, 500, 1000, 2000, 2250, 2500 and 3000 μmol m^−2^ s^−1^ was obtained under ambient CO_2_. Evaluations of gas exchange were made in the morning, between 8:00 and 11:00 h, with fifteen most expanded leaves per plot. During the evaluations, the CO_2_ concentration in the chamber was maintained at 400 µmol mol^−1^.

For the determination of dry biomass (MS) productivity, the aerial part of the plants located in an area of 3 m^2^ in each plot was collected in the flowering phase of the crops. The fresh weight of the plant residues under each water regime was determined, and then the fresh weight of a subsample, which was kept in an oven at 65 °C for 72 h. After this period, the dry weight of the sample was determined. The total dry biomass was determined using the equation: TDB = (PFA × PSA)/PFA. The results were corrected for ton ha^−1^, where:

PST = total dry biomass;

PFT = total fresh weight;

PSA = dry weight of the sample and;

PFA = fresh weight of the sample.

The results were corrected for ton ha^−1^.

Grain productivity was measured by mechanically harvesting all the plants in an area of 3 m^2^ plot^−1^. The common bean was harvested 88 days after planting (DAP), and the other crops were collected 105 DAP with a mechanical harvester. After harvesting, the total grain weight was determined. A subsample was taken to a forced circulation oven, to determine the moisture content of the grains. The ASAE [74] methodology was adopted to determine the moisture in the crop grains (103 °C/72 h) until constant weight. After being removed from the greenhouse, common beans was placed in a desiccator until they reached room temperature. Then, they were weighed on a precision analytical balance. Subsequently, the productivity was corrected to a standard humidity of 13% and the results were expressed in kg ha^−1^.

Productivity per unit of applied water (PUAA) was also evaluated, calculated using the following equation: PUAA = Prod/LTD, in which:

PUAA = productivity per unit of water applied (kg ha mm^−1^);

Prod = grain productivity, in kg ha^−1^;

LTD = total irrigation depth available, referring to the amount of water applied during the crop cycle, in mm.

The temperature of the canopy was determined through thermographic image evaluation of the plants, using a thermal infrared camera (FLIR^®^ T420, FLIR Systems, Wilsonville, OR, USA) with the following characteristics: thermal spatial resolution of 320 × 240 pixels, spectral response from 7.5 to 13 uM, thermal sensitivity of pixels of 0.045 °C to 30 °C and precise temperature ±2 °C, coupled with an unmanned aerial vehicle (XFly, the X800, Bauru, SP, Brazil), at a height of approximately 60 m. Figure 5 shows the RGB images of each species (Figure 5A–D) and the thermal images (Figure 5E–H). The images were taken at the same time and on the same day of the physiological variables of gas exchange. For processing the images and obtaining the temperature data, the QGIS software [75] was used. The RGB (red, green and blue) and thermal images were georeferenced for the generation of orthomosaic, and later classified to select only the canopy areas of the plants in the useful plot, excluding the soil.

### 4.3. Statistical Analysis

The data were initially analysed for normal distribution using the Kolmogorov–Smirnov test. Subsequently, they were subjected to an analysis of variance (ANOVA), and the means were compared via the Tukey test, at 5% probability, using the statistical software SAS, version 9 [76]. The statistical model was adjusted using Proc Mixed from SAS, using the maximum restricted likelihood (reml) method. Pearson’s correlation analysis was performed using the R software [77], and the data were presented using a correlogram.

## 5. Conclusions

In general, water stress negatively affected grain yield and physiological variables; however, to different degrees across the four crops studied. We recommend the cultivation of amaranth and quinoa in the Cerrado areas during the off-season period. They maintained high productivity under the regimes equal to the off-season water regime, a water situation that is common during the off-season period in most of the agricultural regions of the Cerrado.

The use of thermography proved to be an efficient tool for predicting the physiological /productive behaviour of the species studied, as it associated the responses of the physiological and productive variables in water deficit with the increase in the temperature of the canopy.

## Figures and Tables

**Figure 1 plants-12-02081-f001:**
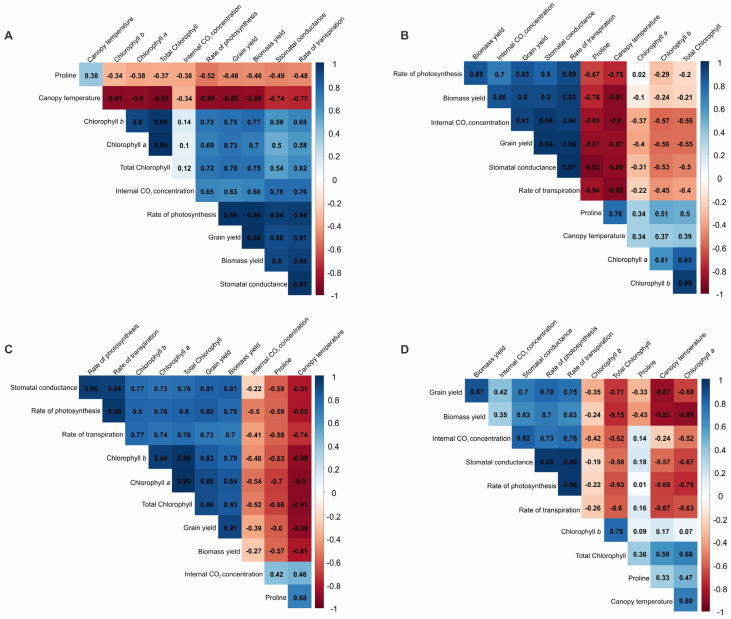
Correlogram of Pearson’s correlation estimates between physiological, productive and temperature variables. (**A**) Common bean; (**B**) quinoa; (**C**) amaranth; (**D**) buckwheat.

**Figure 2 plants-12-02081-f002:**
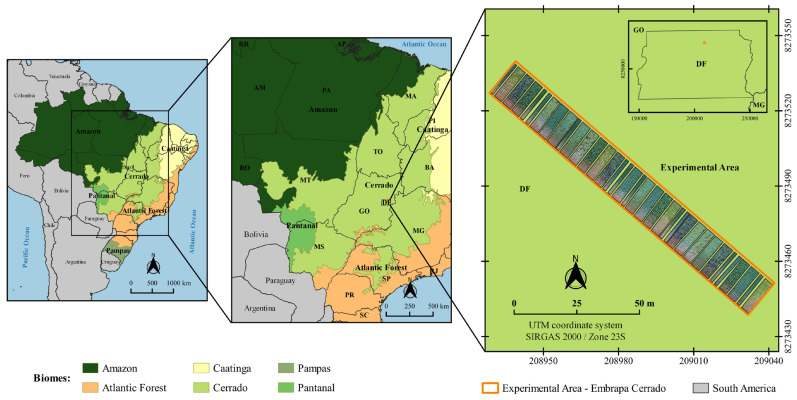
Extension of the Cerrado biome and location of the experimental area located at the Agricultural Research Center of the Cerrado (Embrapa Cerrados), Planaltina, DF, Brazil.

**Figure 3 plants-12-02081-f003:**
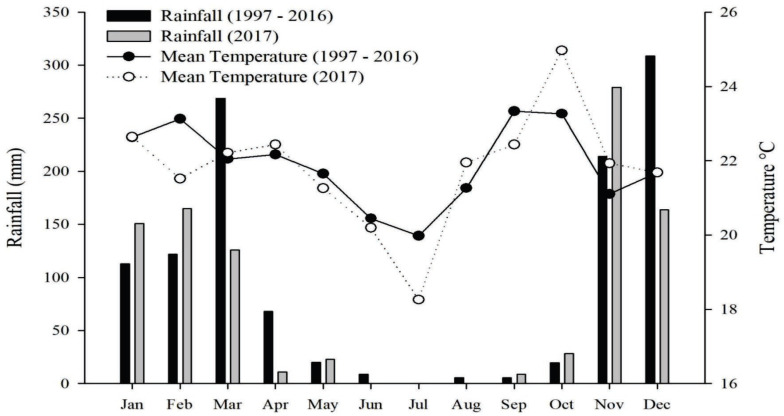
Average rainfall and temperature between 1997 and 2016, and for 2017. The data were obtained from an automatic meteorological station located next to the experiment.

**Figure 4 plants-12-02081-f004:**
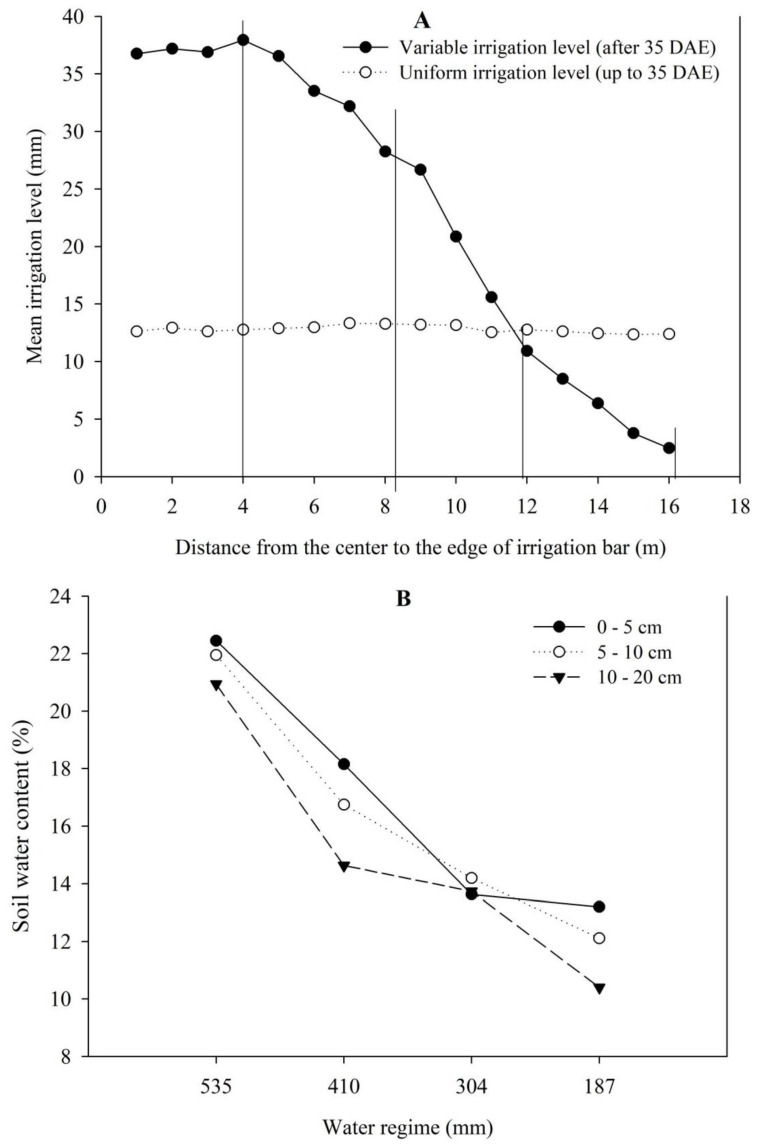
(**A**) Average water depth applied during each period of uniform irrigation with sprinklers with equal flows until 35 DAE, and average water depth applied in each irrigation using sprinklers with decreasing flow rates, from the beginning to the end of the lateral irrigating bar, after 35 DAE. (**B**) Soil humidity at the depths of 0–5, 5–10 and 10–20 cm under the four water regimes applied.

**Figure 5 plants-12-02081-f005:**
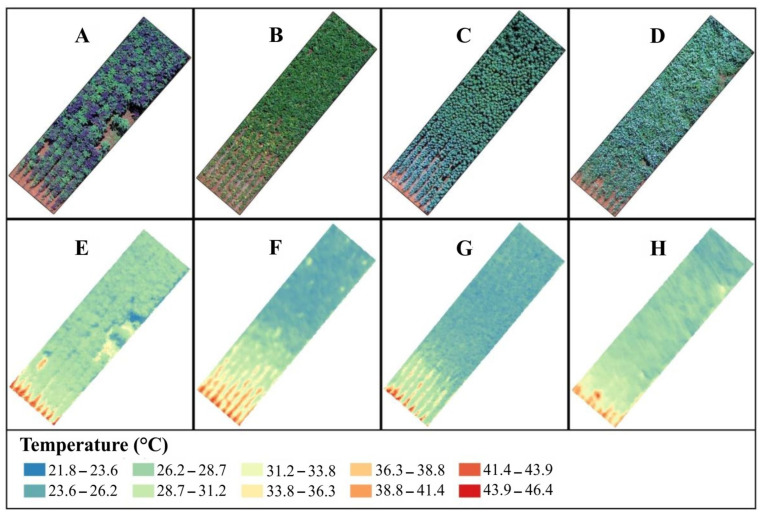
Images of the plots cultivated with (**A**) Amaranthus, (**B**) common bean, (**C**) quinoa and (**D**) buckwheat, and the thermal images of the same plots: (**E**) Amaranthus, (**F**) common bean, (**G**) quinoa and (**H**) buckwheat.

**Table 1 plants-12-02081-t001:** Net photosynthesis (A), stomatal conductance (gs), internal CO_2_ concentration (Ci), transpiration (E) and water use efficiency (WUE) for amaranth, common bean, quinoa and buckwheat crops grown under different water regimes.

Water Regime	Amaranth	Common Bean	Quinoa	Buckwheat
	Photosynthesis—A (µmol CO_2_ m^−2^ s^−1^)			
535	43.86 Aab	25.71 Ca	35.53 Ba	23.87 Ca
410	43.80 Aab	21.98 Ca	35.13 Ba	12.57 Db
304	40.24 Aa	8.47 Bb	30.95 Aa	7.66 Bb
187	25.45 Ab	4.77 Bb	10.90 Bb	4.32 Bb
	Stomatal conductance—gs (µmol CO_2_ m^−2^ s^−1^)			
535	0.30 Da	0.56 Ba	0.73 Aa	0.40 Ca
410	0.31 Ba	0.34 Bb	0.74 Aa	0.10 Cb
304	0.27 ABab	0.06 CDc	0.39 Ab	0.05 Db
187	0.18 Ab	0.04 Cc	0.07 Abc	0.03 Cb
	Internal CO_2_ concentration—Ci (µmol CO_2_ m^−2^ s^−1^)			
535	120.64 Ba	291.43 Aa	283.64 Aa	260.84 Aa
410	128.31 Ba	256.11 Aab	280.72 Aa	152.43 Bb
304	116.55 Ba	200.27 Bbc	223.37 Aa	108.91 Bb
187	142.26 Aa	135.76 BC	124.42 Ab	154.15 Ab
	Transpiration (µmol H_2_O m^−2^ s^−1^)			
535	8.01 Ca	9.93 Ba	12.10 Aa	10.03 Ba
410	7.58 Ba	7.50 Bb	11.89 Aa	4.09 Cb
304	7.56 Aab	2.23 Cc	8.73 Ab	2.82 Cbc
187	5.35 Ab	1.30 Bc	2.53 Bc	1.79 Bc
	Water use efficiency—WUE			
535	5.80 Aa	2.67 Ba	4.30 Ba	2.39 Ba
410	5.43 Aa	2.92 Ba	3.58 Ba	3.08 Ba
304	5.31 Aab	4.17 BCa	2.94 Ca	2.58 Ca
187	4.73 Aab	3.80 BCa	2.92 BCa	2.64 Ca

The means followed by the same capital letters (lines) and lowercase letters (columns) do not differ according to Tukey’s test at 5% probability.

**Table 2 plants-12-02081-t002:** Chlorophyll indexes (*a*, *b* and total) and proline concentration for the cultures of amaranth, common bean, quinoa and buckwheat grown under different water regimes.

Water Regime	Amaranth	Common Bean	Quinoa	Buckwheat
	Chlorophyll *a*			
535	42.01 Aa	39.47 Aa	40.39 Aa	39.77 Aa
410	41.90 Aa	39.90 Aa	43.43 Aa	40.89 Aa
304	40.04 Aa	39.21 Aa	42.94 Aa	41.69 Aa
187	33.39 Bb	34.77 Bb	43.62 Aa	44.18 Aa
	Chlorophyll *b*			
535	9.76 Ba	9.63 Ba	20.20 Aab	12.91 Ba
410	10.61 Ba	9.67 Ba	19.99 Ab	10.91 Ba
304	9.39 Cab	9.20 Ca	23.06 Aab	15.75 Ba
187	4.43 Cb	6.40 Cb	25.18 Aa	13.78 Ba
	Total Chlorophyll			
535	120.64 Ba	291.43 Aa	283.64 Aa	260.84 Aa
410	128.31 Ba	256.11 Aab	280.72 Aa	152.43 Bb
304	116.55 Ba	200.27 Bbc	223.37 Aa	108.91 Bb
187	142.26 Aa	135.76 BC	124.42 Ab	154.15 Ab
	Proline (µmol g^−1^ FM)			
535	0.128 Ab	0.123 Ab	0.106 Ab	0.084 Aa
410	0.155 Aab	0.129 Ab	0.116 Ab	0.064 Aa
304	0.196ABab	0.185 Ba	0.258 Aa	0.058 Ca
187	0.261 Aba	0.189 Ba	0.314 Aa	0.097 Ca

The means followed by the same uppercase letters in the lines and lowercase letters in the columns do not differ according to Tukey’s test at the 5% probability level. FM: fresh leaf mass.

**Table 3 plants-12-02081-t003:** Dry biomass production, grain yield and productivity per unit of water applied (PUAD in kg ha mm^−1^) for amaranth, common bean, quinoa and buckwheat grown under different water regimes.

Water Regime (mm)	Amaranth	Common Bean	Quinoa	Buckwheat
Biomass production (kg ha^−1^)				
535	16,669.85 Aa	6650.88 Ca	14,756.53 Aa	8513.8 Ba
410	15,592.47 Aa	6590.66 Ca	18,289.1 Aa	10,820.33 Ba
304	9449.5 Bb	4599.46 Bb	15,881.52 Aa	8289.08 Ba
187	3490.91 Ac	1162.09 Bc	4878.83 Ab	4533.17 Ab
Grain productivity (kg ha^−1^)				
535	3450.01 Ca	5295.47 Aa	4084.69 Ba	2387.43 Da
410	3575.01 Ba	4383.21 Aa	3622.80 Ba	2095.95 Ca
304	2724.17 Ba	1613.63 Cb	3266.17 Aa	1892.40 Cb
187	567.58 Bb	465.25 Bc	541.02 Bb	725.29 Ac
Productivity per unit of water applied (kg ha mm^−1^)				
535	6.51 BCa	9.89 Aab	7.90 ABb	4.46 Ca
410	8.68 Aa	10.69 Aa	8.42 Aab	5.11 Ba
304	9.00 Aba	7.83 BCb	10.88 Aa	6.22 Ca
187	2.92 Ab	3.99 Ac	3.14 Ac	3.87 Aa

The means followed by the same lowercase (columns) and uppercase (lines) letters do not differ according to Tukey’s test at 5% probability.

**Table 4 plants-12-02081-t004:** Average temperature values (°C) of the canopy of amaranth, common bean, quinoa and buckwheat under the four water regimes studied.

Water Regime	Amaranth	Common Bean	Quinoa	Buckwheat
535	27.05 Ab	25.31 Abc	25.01 Cb	26.04 Bb
410	26.90 Ab	25.25 Bc	24.92 Bb	26.74 Ab
304	27.29 Bb	28.50 Ab	25.66 Cb	27.01 Bb
187	31.89 Ba	34.54 Aa	29.79 Ca	30.12 Ca

The means followed by the same lowercase (columns) and uppercase (lines) letters do not differ according to Tukey’s test at 5% probability.

**Table 5 plants-12-02081-t005:** Description of the cultivation history of the study area in the period between 2005 and 2016.

Harvest	Period	
	Winter	Summer
2005/2006	Fallow	Soybean
2006/2007	Fallow	Soybean
2007/2008	Fallow	Soybean
2008/2009	Fallow	Soybean
2009/2010	Fallow	Soybean
2010/2011	Fallow	Soybean
2011/2012	Soybean under different water regimes	Fallow
2012/2013	Wheat under different water regimes	Soybean
2013/2014	*A. cruenthus*, *P. glaucum* and *C. quinoa* under different water regimes	*Crotalaria juncea*
2014/2015	*A. cruenthus*, *P. glaucum* and *C. quinoa* under different water regimes	*Zea mays*
2015/2016	*A. cruenthus*, *P. glaucum* and *C. quinoa* under different water regimes	*Crotalaria juncea*

**Table 6 plants-12-02081-t006:** Morphological, productive characteristics and photosynthesis mechanism of the studied species.

Culture	Features
Common bean (Carioca)	Determined growth habit (Type I), erect posture, emergence cycle to physiological maturation of approximately 67 days. It has an average yield of 1893 kg ha^−1^ in the water harvest, 2174 kg ha^−1^ in the dry season and 2269 kg ha^−1^ in the winter. It has a C_3_ photosynthesis mechanism.
Amaranth (BRS Alegria)	Average height of 1.8 m; period between emergence and physiological maturation is 90 days. The average grain yield is 2359 kg ha^−1^. It can be cultivated at any time of the year: for grain production off-season and winter, cultivation is recommended, whilst for forage production summer, sowing is ideal. Has a C_4_ photosynthesis mechanism.
Quinoa (BRS Piabiru)	Average height of 1.9 m; period between emergence and physiological maturity is 110 days; average productivity is 2800 kg ha^−1^. It can be cultivated at any time of the year: for grain production off-season and winter, cultivation is recommended, whilst for forage production, it can be sown at the beginning of the rainy season. It has a C_3_ photosynthesis mechanism.
Buckwheat (IPR 91)	Upright shrub growth habit; average height of 1.5 to 1.8 m; production of 3 to 6 tons per hectare of dry biomass and 15 to 25 tons per hectare of fresh biomass. For grain production, it can be planted from October to December (recommended) or January to March. It has a C_3_ photosynthesis mechanism.

## Data Availability

The data presented in this study are available from the corresponding author upon reasonable request.

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
