# Peer review of "Use of Thermography to Evaluate Alternative Crops for Off-Season in the Cerrado Region"

_plants, 2023, doi:10.3390/plants12112081_

Round 1

Reviewer 1 Report

The manuscript evaluated the effect of water stress on the physiology and productivity of four species of crops in off-season with the designed experimental and plenty of data. The results of two potential species were recommended were very important with reference value for cultivation in off-season in Cerrado region. However, there are some parts on analysis and background introduction in this manuscript, which could be improved to make the manuscript better understanding.

The main issue is the content of this manuscript was not quite relevant to the title. The emphasis of the title is “Use of thermography”, however, the manuscript spent much of the results analyzing physiology and productivity of crops, not how thermography could reflect the morphophysiological effect. Therefore, it is suggested that the authors make some modification logically.

(1)    the relationship between canopy temperature and physiology of crops should be added as groundwork in the INTRODUCTION part. Then the hypothesis of that water stress is correlated with the temperature of the canopy of crops could be made. More relative research results should be mentioned here.

(2)    The results of canopy temperature from thermal images should be given at first in RESULTS part. Thereafter the physiological variables and productivity of four species in different water stress, followed by the correlation between canopy temperature and other index.

Author should also check the obvious mistakes in the manuscript, like:

In Line 415, the precipitation and temperature data are shown in Figure 3, not in Figure 2.

In Line 439, table 6 cannot be found.

In Line 547, Figure 5 shows the RGB image, not Figure 4.

Author Response

Reviewer 1

The manuscript evaluated the effect of water stress on the physiology and productivity of four species of crops in off-season with the designed experimental and plenty of data. The results of two potential species were recommended were very important with reference value for cultivation in off-season in Cerrado region. However, there are some parts on analysis and background introduction in this manuscript, which could be improved to make the manuscript better understanding.

Authors

We corrected and included more references related to canopy temperature in the introduction

Reviewer

The main issue is the content of this manuscript was not quite relevant to the title. The emphasis of the title is “Use of thermography”, however, the manuscript spent much of the results analyzing physiology and productivity of crops, not how thermography could reflect the morphophysiological effect. Therefore, it is suggested that the authors make some modification logically.

(1)    the relationship between canopy temperature and physiology of crops should be added as groundwork in the INTRODUCTION part. Then the hypothesis of that water stress is correlated with the temperature of the canopy of crops could be made. More relative research results should be mentioned here.

Authors

We corrected this part and even altered the title of the paper in the introduction. in the discussion, we corrected and improved the discussion of the data.

Reviewer

(2)    The results of canopy temperature from thermal images should be given at first in RESULTS part. Thereafter the physiological variables and productivity of four species in different water stress, followed by the correlation between canopy temperature and other index.

Authors

The order of the discussion was not changed, as we are using the thermal evaluation (canopy temperature) at the end, because we need previous results, such as gas exchange, chorophyll and Other parameters to be used for validating the thermography data.  

Reviewer

Author should also check the obvious mistakes in the manuscript, like:

Reviewer

In Line 415, the precipitation and temperature data are shown in Figure 3, not in Figure 2.

Authors – We corrected

Reviewer

In Line 439, table 6 cannot be found.

Reviewer

Authors – we included, sorry about this mistake.

Reviewer

In Line 547, Figure 5 shows the RGB image, not Figure 4.

Authors – we corrected

Reviewer 2 Report

This paper describes the use of thermography to monitor and evaluate the state of crops under different water regimes. This application is well known but this work represents a further validation of the thermography contribute to the precision agriculture.

The work is well detailed and complete and the authors report a huge quantity of experimental results, well analyzed in the Discussion section, regarding the behavior of the studied crops according to the photosynthetic rate under each considered water regime. Nevertheless, in the Results session (lines 253-259) it seems necessary to give an interpretation about the different correlations concerning the average canopy temperatures.

Author Response

Reviewer 2.

This paper describes the use of thermography to monitor and evaluate the state of crops under different water regimes. This application is well known but this work represents a further validation of the thermography contribute to the precision agriculture.

The work is well detailed and complete and the authors report a huge quantity of experimental results, well analyzed in the Discussion section, regarding the behavior of the studied crops according to the photosynthetic rate under each considered water regime. Nevertheless, in the Results session (lines 253-259) it seems necessary to give an interpretation about the different correlations concerning the average canopy temperatures.

Authors

We included more discussion about this issue along the paper.

Round 2

Reviewer 1 Report

The author modified the title, imporved the introduction and corrected the pointed error, which resulted in the manuscript clear to understand.

Line 398, modify stomaltal to stomatal.